health and disease and epidemiology/genetics

COVID-19, Mendelian randomization, angiotensin-converting enzyme inhibitors, genetic epidemiology

**Author for correspondence:**
Stephen Burgess
e-mail: sb452@medschl.cam.ac.uk

[†]Deceased.
[‡]Present address: MRC Biostatistics Unit, Cambridge Institute of Public Health, Robinson Way, Cambridge, CB2 0SR, UK.

# ACE inhibition and cardiometabolic risk factors, lung *ACE2* and *TMPRSS2* gene expression, and plasma ACE2 levels: a Mendelian randomization study

Dipender Gill[1], Marios Arvanitis[2], Paul Carter[5],
Ana I. Hernández Cordero[12], Brian Jo[13], Ville Karhunen[1],
Susanna C. Larsson[14,15], Xuan Li[12], Sam M. Lockhart[6],
Amy Mason[7,16], Evanthia Pashos[17], Ashis Saha[3],
Vanessa Y. Tan[18,19], Verena Zuber[1,8], Yohan Bossé[20],
Sarah Fahle[7,9,16], Ke Hao[21], Tao Jiang[7],
Philippe Joubert[20], Alan C. Lunt[1,†],
Willem Hendrik Ouwehand[22,23,24], David J. Roberts[9,25,26],
Wim Timens[27], Maarten van den Berge[28],
Nicholas A. Watkins[9,23], Alexis Battle[4],
Adam S. Butterworth[7,9,10,16,29], John Danesh[7,9,10,16,24,29],
Emanuele Di Angelantonio[7,9,10,16,23,29],
Barbara E. Engelhardt[30], James E. Peters[29,31],
Don D. Sin[12] and Stephen Burgess[7,8,10,11,16,‡]

[1]Department of Epidemiology and Biostatistics, St Mary's Hospital, Imperial College London, Medical School Building, London, UK
[2]Department of Medicine, Division of Cardiology, [3]Department of Computer Science, and [4]Department of Biomedical Engineering and Center for Computational Biology, Johns Hopkins University, Baltimore, MD, USA
[5]Department of Public Health and Primary Care, [6]Medical Research Council Metabolic Diseases Unit, Wellcome Trust-Medical Research Council Institute of Metabolic Science, [7]British Heart Foundation Cardiovascular Epidemiology Unit, Department of Public Health

and Primary Care, [8]Medical Research Council Biostatistics Unit, Cambridge Institute of Public Health, [9]National Institute for Health Research Blood and Transplant Research Unit in Donor Health and Genomics, [10]British Heart Foundation Centre of Research Excellence, and [11]Homerton College, University of Cambridge, Cambridge, UK

[12]The University of British Columbia Centre for Heart Lung Innovation, St Paul's Hospital, Vancouver, BC, Canada

[13]Program in Quantitative and Computational Biology, Lewis Sigler Institute for Integrative Biology, Princeton, NJ, USA

[14]Unit of Cardiovascular and Nutritional Epidemiology, Institute of Environmental Medicine, Karolinska Institutet, Stockholm, Sweden

[15]Department of Surgical Sciences, Uppsala University, Uppsala, Sweden

[16]National Institute for Health Research Cambridge Biomedical Research Centre, University of Cambridge and Cambridge University Hospitals, Cambridge, UK

[17]Internal Medicine Research Unit, Pfizer Worldwide Research, Development & Medical, Cambridge, MA, USA

[18]Medical Research Council Integrative Epidemiology Unit, University of Bristol, Bristol, UK

[19]Population Health Sciences, Bristol Medical School, University of Bristol, Bristol, UK

[20]Institut universitaire de cardiologie et de pneumologie de Québec – Université Laval, Quebec, Canada

[21]Department of Genetics and Genomic Sciences, Icahn Institute for Data Science and Genomic Technology, Icahn School of Medicine at Mount Sinai, New York, NY, USA

[22]Department of Haematology, University of Cambridge, Cambridge Biomedical Campus, Cambridge, UK

[23]NHS Blood and Transplant, Cambridge Biomedical Campus, Cambridge, UK

[24]Wellcome Sanger Institute, Cambridge, UK

[25]NHS Blood and Transplant-Oxford Centre, Level 2, John Radcliffe Hospital, Oxford, UK

[26]Radcliffe Department of Medicine, University of Oxford, John Radcliffe Hospital, Oxford, UK

[27]Department of Pathology and Medical Biology and Groningen Research Institute for Asthma and COPD, and [28]Department of Pulmonology and Groningen Research Institute for Asthma and COPD, University of Groningen, Groningen, The Netherlands

[29]Health Data Research UK Cambridge, Wellcome Genome Campus and University of Cambridge, Cambridge, UK

[30]Computer Science Department and Center for Statistics and Machine Learning, Princeton University, Princeton, NJ, USA

[31]Department of Immunology and Inflammation, Faculty of Medicine, Imperial College London, London, UK

YB, 0000-0002-3067-3711; SF, 0000-0002-4359-9952; SB, 0000-0001-5365-8760

Angiotensin-converting enzyme 2 (ACE2) and serine protease TMPRSS2 have been implicated in cell entry for severe acute respiratory syndrome coronavirus 2 (SARS-CoV-2), the virus responsible for coronavirus disease 2019 (COVID-19). The expression of *ACE2* and *TMPRSS2* in the lung epithelium might have implications for the risk of SARS-CoV-2 infection and severity of COVID-19. We use human genetic variants that proxy angiotensin-converting enzyme (ACE) inhibitor drug effects and cardiovascular risk factors to investigate whether these exposures affect lung *ACE2* and *TMPRSS2* gene expression and circulating ACE2 levels. We observed no consistent evidence of an association of genetically predicted serum ACE levels with any of our outcomes. There was weak evidence for an association of genetically predicted serum ACE levels with *ACE2* gene expression in the Lung eQTL Consortium ($p = 0.014$), but this finding did not replicate. There was evidence of a positive association of genetic liability to type 2 diabetes mellitus with lung *ACE2* gene expression in the Gene-Tissue Expression (GTEx) study ($p = 4 \times 10^{-4}$) and with circulating plasma ACE2 levels in the INTERVAL study ($p = 0.03$), but not with lung *ACE2* expression in the Lung eQTL Consortium study ($p = 0.68$). There were no associations of genetically proxied liability to the other cardiometabolic traits with any outcome. This study does not provide consistent evidence to support an effect of serum ACE levels (as a proxy for ACE inhibitors) or cardiometabolic risk factors on lung *ACE2* and *TMPRSS2* expression or plasma ACE2 levels.

# 1. Introduction

Severe acute respiratory syndrome coronavirus 2 (SARS-CoV-2) is responsible for the current coronavirus disease 2019 (COVID-19) pandemic [1]. Serine protease TMPRSS2 is involved in priming the SARS-CoV-2 spike protein for cellular entry through the angiotensin-converting enzyme 2 (ACE2) receptor [2–5]. COVID-19 patients most frequently present with respiratory tract infection symptoms [6–11]. It follows that the expression of *ACE2* and *TMPRSS2* in the lung epithelium might have implications for the risk of SARS-CoV-2 infection and severity of COVID-19 [2,12,13].

Emerging evidence suggests that patients with underlying cardiometabolic risk factors and airway disease are more likely to suffer from severe COVID-19 [6–11]. It has been speculated that the angiotensin-converting enzyme inhibitor (ACEi) and angiotensin receptor blocker (ARB) classes of antihypertensive medication that are more commonly prescribed in patients with cardiometabolic risk factors might affect

the expression of ACE2 and thus affect susceptibility to SARS-CoV-2 infection and severity of consequent COVID-19 [14–20]. Although ACE and ACE2 are both dipeptidyl carboxydipeptidases, they have distinct physiological effects. ACE cleaves angiotensin I to angiotensin II, which consequently activates the angiotensin II receptor type 1 pathway resulting in vasoconstriction and inflammation. By contrast, ACE2 degrades angiotensin II to angiotensin 1–7 and angiotensin I to angiotensin 1–9. Angiotensin 1–9 activates the Mas receptor to have vasodilatory and anti-inflammatory effects. Animal studies have supported effects of ACEi and ARB drugs on ACE2 expression and activity [21–28], with mixed findings for associations of ACEi and ARB drug use with ACE2 activity and levels in human tissues also reported [29–31]. It is important that any causal effects of these medications and cardiometabolic traits on ACE2 expression be further investigated. Identification of a mechanistic basis by which such exposures affect the risk and severity of COVID-19 could provide useful insight for disease prevention and treatment. This could be used to inform optimal medication use and strategies for shielding vulnerable individuals, as well as improving the evidence base for public health campaigns.

The Mendelian randomization approach uses genetic variants related to exposure as instrumental variables for investigating the effect of that exposure on an outcome [32]. Genetic variants are treated analogously to treatment allocation in a randomized controlled trial. Typically, molecular measurements such as gene expression or circulating protein levels are regarded in Mendelian randomization investigations as exposure variables. Here, following the work of Rao *et al*. [33], we treat these molecular measurements as the outcomes in our investigation. The aim of this study was to apply Mendelian randomization to investigate whether ACE2 and TMPRSS2 gene expression in the lung and circulating levels of ACE2 in the plasma are associated with (i) genetic variants in the ACE gene region that can be considered as proxies for the effect of ACEi drugs and (ii) genetic variants related to cardiometabolic risk factors. We also use publicly available data on genetic associations with COVID-19 susceptibility to investigate whether genetically predicted serum ACE levels are associated with risk of hospitalization due to COVID-19.

# 2. Methods

## 2.1. Genetic associations with exposure variables

Two different genetic instruments that proxy ACEi drug effects were considered. First, we selected 17 single-nucleotide polymorphisms (SNPs) in the ACE locus that were associated with serum ACE concentration in the Outcome Reduction with Initial Glargine INtervention (ORIGIN) trial and did not have strong pairwise correlation ($r^2 < 0.1$) [34]. Accounting for correlation, these variants explain 29.0% of the variance in serum ACE concentration, corresponding to an F statistic of 85.4 (INTERVAL), 17.7 (Lung eQTL Consortium) and 8.8 (GTEx). Second, we selected a single SNP, rs4291, located at the ACE locus that was associated with systolic blood pressure (SBP) at $p = 9 \times 10^{-20}$ in a study of 757 601 European-ancestry individuals [35,36]. Each blood pressure-lowering allele of this SNP was associated with a 0.28 mmHg reduction in SBP [35]. This variant explains <0.1% of the variance in blood pressure. There were no other SNPs at this locus that were associated with SBP at a genome-wide level of significance ($p < 5 \times 10^{-8}$) and did not have strong pairwise correlation ($r^2 < 0.1$) with the index variant [35]. To assess the validity of the serum ACE variants, we assessed their associations with SBP in the UK Biobank study [37].

We further considered six cardiometabolic traits as exposure variables: body mass index (BMI), chronic obstructive pulmonary disease (COPD), lifetime smoking index, low-density lipoprotein cholesterol (LDL-C), SBP and type 2 diabetes mellitus (T2DM). These traits were chosen as they have been associated with prognosis of COVID-19 [6–11]. Genetic association estimates for these exposures were obtained from the publicly available genome-wide association study (GWAS) summary data sources listed in table 1. Genetic variants selected as instruments were SNPs associated with the corresponding trait at a genome-wide level of statistical significance ($p < 5 \times 10^{-8}$) and were uncorrelated ($r^2 < 0.001$). Clumping of correlated variants was performed using the TwoSampleMR package in R [43].

## 2.2. Genetic associations with outcome variables

Genetic associations with the expression of ACE2 and TMPRSS2 in lung tissue were obtained from two sources: (i) the Gene-Tissue Expression (GTEx) project [44] and (ii) the Lung eQTL (expression quantitative trait loci) Consortium [45]. Genetic associations with risk of hospitalization due to COVID-19 were obtained from release 4 (alpha version) of the COVID-19 Host Genomics Initiative [46].

**Table 1.** Sources for exposure trait genome-wide association study summary data.

| trait | sample size | population ancestry | number of variants | variance explained (%) | reference |
|---|---|---|---|---|---|
| body mass index (BMI) | 806 834 | European | 546 | 5.7 | [38] |
| chronic obstructive pulmonary disease (COPD) | 35 735 cases and 222 076 controls | Predominantly European | 82 | 7.0 | [39] |
| lifetime smoking index | 462 690 | European | 126 | 0.36 | [40] |
| low-density lipoprotein cholesterol (LDL-C) | 188 577 | European | 80 | 7.9 | [41] |
| systolic blood pressure (SBP) | 318 417 | British | 192 | 2.9 | [37] |
| type 2 diabetes mellitus (T2DM) | 74 124 cases and 824 006 controls | European | 202 | 16.3 | [42] |

GTEx genetic association estimates were obtained in 515 individuals of predominantly European (85%) ancestry. RNA sequencing was performed using the Illumina TruSeqTM RNA sample preparation protocol, and gene-level expression quantification was performed using RNA-SeQC for gene-level read counts and transcripts per million values [47]. Whole-genome sequencing was performed by the Broad Institute's Genomics Platform and only common variants (minor allele frequency >0.05) were retained. Genome-wide eQTL analysis was performed for the expression of *ACE2* and *TMPRSS2* in primary tissue samples taken from the lung. Genetic associations with imputed variants across the autosomal chromosomes were adjusted for five principal components from the genotype data, 60 probabilistic estimations of expression residuals factors [48], sequencing platform (Illumina HiSeq 2000 or HiSeq X), sequencing protocol (polymerase chain reaction-based or free) and sex. For each gene, expression values between samples were normalized using the trimmed means of M-values method in EdgeR [49]. Expression values were normalized across samples using an inverse-normal transformation.

Lung eQTL Consortium genetic association estimates were obtained in 1038 individuals of European ancestry [45]. Tissue samples were obtained at three different institutions: University of British Columbia, Laval University and University of Groningen. Genome-wide eQTL analysis was performed for the expression of *ACE2* using a probe set 100134205_TGI_at and two probe sets for *TMPRSS2*, 100130004_TGI_at and 100157336_TGI_at (subsequently referred to as 1 and 2). Expression profiling was performed using an Affymetrix custom array (see Gene Expression Omnibus platform GPL10379) [45]. The probe sets measure different transcripts, and the specific probes for each probe set are detailed in electronic supplementary material, table S1. Expression levels were much higher for the first probe set, and so these results are more reliable. All participants were genotyped using the Illumina Human 1 M Duo BeadChip and the genotypes were imputed using the Haplotype Reference Consortium reference panel. Expression values were first standardized for age, sex and smoking status using robust linear regression. Genetic associations were estimated in each cohort separately using a linear additive genetic model. The estimates were combined across cohorts using an inverse-variance weighted model with fixed effects.

Genetic association estimates with circulating plasma ACE2 levels were obtained in a subcohort of 4998 blood donors enrolled in the INTERVAL BioResource [50]. Plasma ACE2 levels were measured using a multiplex proximity extension immunoassay (Cardiovascular 2 panel, Olink Bioscience, Uppsala, Sweden). A total of 4947 samples passed quality control. The data were pre-processed using standard Olink workflows including applying median centring normalization across plates, where the median is centred to the overall median for all plates, followed by $\log_2$ transformation to provide normalized protein levels (NPX). NPX values were regressed on age, sex, plate, time from blood draw to processing (in days) and season. The residuals were then rank-inverse normalized. Genotype data were processed as described previously [51]. Genome-wide pQTL analysis was performed by linear regression of the rank-inverse normalized residuals on genotype in SNPTEST [52], with the first three components of multi-dimensional scaling as covariates to adjust for ancestry.

Genetic association estimates with hospitalization due to COVID-19 were obtained from version 4 (release alpha) of the COVID-19 Host Genomics Initiative, which included 6492 cases and 1 012 809 controls from the general population from 17 studies, mostly from participants of European ancestries [46]. Association estimates were obtained within each study with adjustment for age, age-squared, sex, at least 20 principal components and technical covariates, and then meta-analysed across studies.

## 2.3. Mendelian randomization analyses

For the analysis investigating genetically proxied ACEi drug effects using the rs4291 variant, we report the genetic associations with lung *ACE2* and *TMPRSS2* expression, plasma ACE2 levels and risk of COVID-19 hospitalization per blood pressure-lowering allele.

For all other Mendelian randomization analyses, estimates were obtained from the inverse-variance weighted method under a random-effects model [53]. For the polygenic analyses based on the *ACE* gene locus, we accounted for the correlation between variants using generalized weighted regression [54]. Heterogeneity between Mendelian randomization estimates from different genetic variants for the same exposure trait was expressed using the $I^2$ statistic [55]. For any identified associations at $p < 0.05$, the weighted median [56], MR-Egger [57] and contamination-mixture methods [58], which are more robust to the inclusion of pleiotropic variants, were performed as sensitivity analyses.

Mendelian randomization estimates represent the change in the outcome per one standard deviation increase in genetically predicted levels of the exposure for continuous exposure traits and per unit increase in the $\log_e$ odds of the exposure for binary traits. All outcome measures were rank-based inverse-normal transformed, and so changes in the outcome measures are in standard deviation units, with the exception of COVID-19 hospitalization, for which estimates represent odds ratios.

## 2.4. Ethical approval, data availability and reporting

The data used in this work are obtained from published studies that obtained relevant participant consent and ethical approval. All variants used as instruments and their genetic association estimates were selected from publicly available data sources, and are provided in electronic supplementary material, tables S2–S9. GWAS summary data for all the outcomes considered in this study are publicly available at http://dx.doi.org/10.6084/m9.figshare.12102681 (GTEx), http://dx.doi.org/10.6084/m9.figshare.12102711 (Lung eQTL Consortium), and http://dx.doi.org/10.6084/m9.figshare.12102777 (INTERVAL). The results from the analyses performed in this work are presented in the main manuscript or its supplementary files. This paper has been reported based on recommendations by the STROBE-MR Guidelines (Research Checklist) [59]. The study protocol and details were not pre-registered.

# 3. Results

Results for the analyses investigating genetically proxied ACEi drug effects are displayed in figure 1. Genetic associations of the variants with serum ACE levels, SBP and the molecular outcome measures are plotted in electronic supplementary material, figure S1. The variants were associated with SBP in the expected direction: 0.22 mmHg (95% confidence interval 0.06 to 0.37, $p = 0.006$) increase per one standard deviation increase in ACE. There was evidence of an association with *ACE2* expression in the Lung eQTL Consortium for the variants associated with serum ACE: -0.087 standard deviation change (95% confidence interval -0.156 to -0.018, $p = 0.014$) in *ACE2* expression per one standard deviation increase in serum ACE. For the other molecular outcome measures, there was no evidence of associations considering the variants associated with serum ACE (figure 1*a*), or the variant associated with SBP (figure 1*b*). Results were similar in sensitivity analyses restricted to 12 variants associated with serum ACE at a genome-wide level of significance ($p < 5 \times 10^{-8}$) and for the lead variant (rs4343) only (electronic supplementary material, figure S2). There was no evidence of the association between genetically predicted serum ACE and risk of hospitalization due to COVID-19 for the 17 variants (odds ratio 1.02 per standard deviation increase in ACE; 95% confidence interval 0.94 to 1.10) or for the SBP variant (odds ratio 1.05 per blood pressure decreasing allele; 95% confidence interval 0.99 to 1.12).

The main inverse-variance weighted method Mendelian randomization results for the cardiometabolic risk factors are displayed in figure 2 for lung *ACE2* expression and plasma ACE2 concentrations, and in figure 3 for lung *TMPRSS2* expression. There was evidence of a positive association of genetic liability to T2DM with lung *ACE2* gene expression in GTEx ($p = 4 \times 10^{-4}$) and

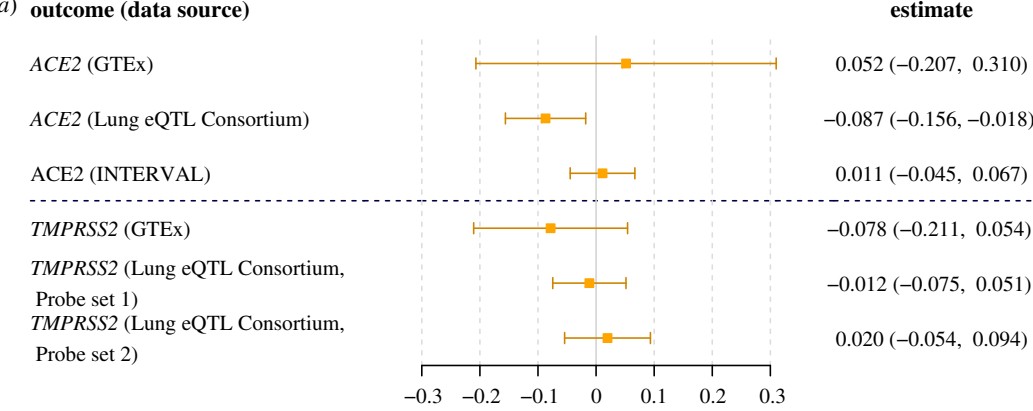

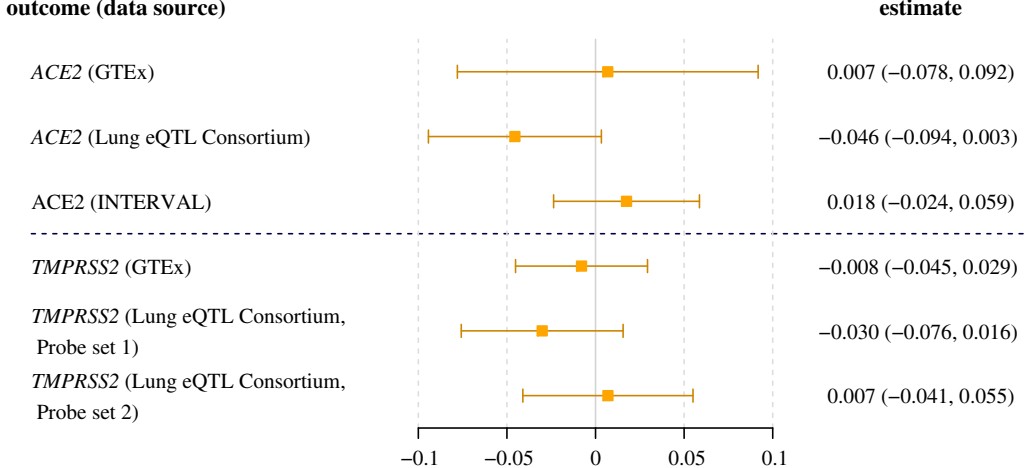

**Figure 1.** Genetic associations with *ACE2* and *TMPRSS2* gene expression in the lung (GTEx and Lung eQTL Consortium) and circulating ACE2 protein levels in the plasma (INTERVAL): (*a*) per one standard deviation increased ACE concentration conferred through variants at the *ACE* gene and (*b*) per blood pressure-lowering allele for the rs4291 variant in the *ACE* gene (bottom panel). The two sets of results for *TMPRSS2* expression in the Lung eQTL consortium refer to two separate probe sets for estimating gene expression.

with circulating plasma ACE2 levels in INTERVAL ($p = 0.03$) (electronic supplementary material, figure S3). Similar point estimates were obtained when performing the weighted median, MR-Egger and contamination-mixture Mendelian randomization sensitivity analyses that are more robust to the presence of pleiotropic variants, although the confidence intervals were wider (electronic supplementary material, table S10). The MR-Egger method did not identify any evidence of directional pleiotropy biasing the analysis (electronic supplementary material, table S10). There was no evidence of an association of genetic liability to T2DM with lung *ACE2* gene expression in the Lung eQTL Consortium ($p = 0.68$). There was no evidence of an association between genetically predicted levels of any of the other cardiometabolic traits with *ACE2* or *TMPRSS2* gene expression in GTEx or the Lung eQTL Consortium, or with circulating plasma ACE2 levels in INTERVAL.

## 4. Discussion

In the current COVID-19 pandemic, there is an urgent need to elucidate mechanisms underlying risk and severity of COVID-19, with a view to informing preventative and therapeutic strategies. In this study, we used human genetic variants that proxy ACEi drug effects and cardiometabolic risk factors to provide insight into how these exposures affect lung *ACE2* and *TMPRSS2* expression and circulating ACE2 levels.

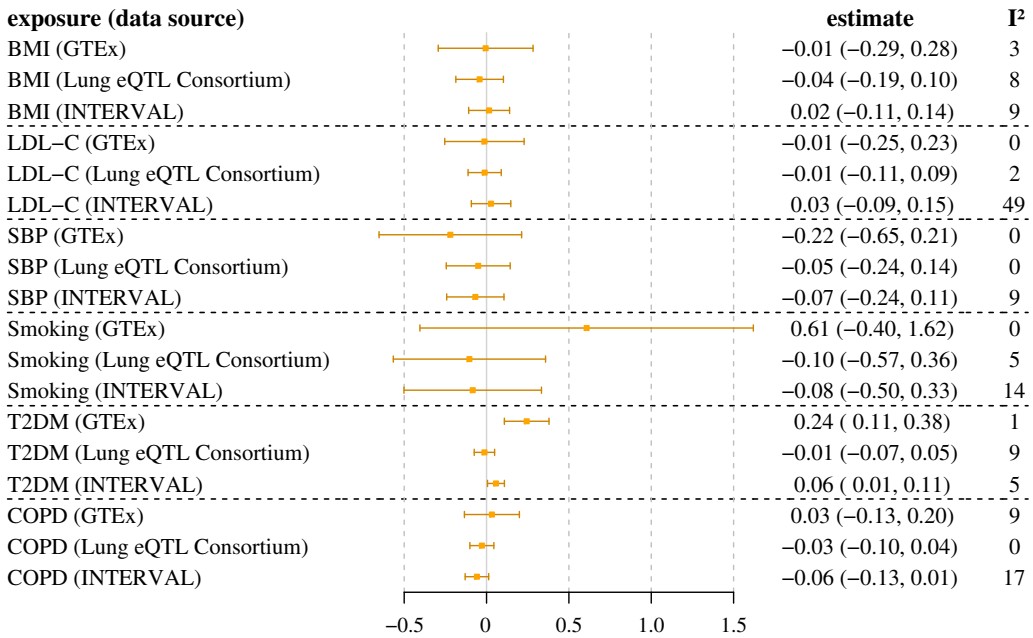

**Figure 2.** Mendelian randomization estimates for the change in *ACE2* gene expression in the lung (GTEx and Lung eQTL Consortium) and circulating ACE2 protein levels in the plasma (INTERVAL) per unit increase in genetically predicted levels of the exposure.

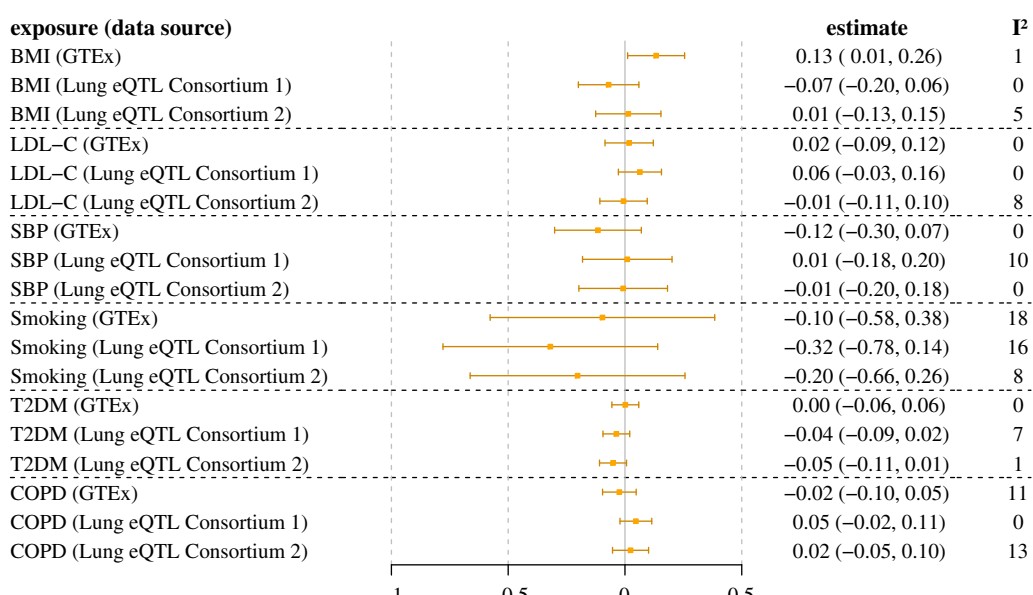

**Figure 3.** Mendelian randomization estimates for the change in *TMPRSS2* gene expression in the lung per unit increase in genetically predicted levels of the exposure. The two sets of results for the Lung eQTL Consortium refer to two separate probe sets for estimating gene expression.

We did not find a consistent association of genetically predicted serum ACE levels with lung *ACE2* and *TMPRSS2* expression or with circulating plasma levels of ACE2. In one dataset, we found evidence that increased serum ACE may lead to decreased expression of *ACE2*, meaning that ACE inhibition would increase *ACE2* expression. However, this finding did not replicate. Additionally, genetically predicted serum ACE levels were not associated with risk of hospitalization due to COVID-19. Previously identified changes in ACE2 expression in human tissues following ACEi treatment may not be applicable to the lung or circulating plasma levels [29,30]. Our findings support the stance of

professional bodies for supporting the continuation of ACEi and ARB antihypertensive drugs in patients with COVID-19 unless there is a clinical justification for stopping [60,61]. Indeed, appropriate use of these medications is of proven benefit [62,63], and their abrupt interruption can also do considerable harm [64,65]. While there has also been speculation that ACEi and ARB antihypertensive drugs might reduce the severity of COVID-19 [19,66–68], with clinical trials to explore this currently planned [60], our findings are also consistent with guidance that patients should not start taking these drug classes unless clinically indicated [60].

Our results identified inconsistent support for an effect of liability to T2DM on lung *ACE2* expression and plasma ACE2 levels. An association of genetic liability to T2DM with lung *ACE2* expression in the GTEx project has previously been described [33]. However, we identified no association of genetic liability to T2DM with lung *ACE2* expression in the Lung eQTL Consortium. This discrepancy may be attributable to the different populations considered in the GTEx project and the Lung eQTL Consortium [44,45]. While the Lung eQTL Consortium considered lung tissue from patients requiring resectional surgery, all samples in the GTEx project were taken from healthy tissue in deceased donors [44,45]. Taken together, our results do not provide consistent support for an effect of cardiometabolic traits on lung *ACE2* or *TMPRSS2* expression or plasma ACE2 levels. While the association between cardiometabolic traits and severity of COVID-19 could be attributable to alternative mechanisms [6–11], these risk factors can still be used to stratify patients in terms of their vulnerability. Similarly, while our current findings do not support a causal effect of COPD or smoking on lung *ACE2* expression, these factors may still be used to inform risk models for severe COVID-19 [69].

Since the initial submission of this manuscript, further data have become available on genetic associations with susceptibility to COVID-19 [46,70]. Mendelian randomization analyses have supported the effect of higher BMI and lifetime smoking on increasing susceptibility to severe COVID-19 [71]. The discrepancy with our current findings, which did not identify an association of genetically predicted BMI or smoking with lung *ACE2* and *TMPRSS2* expression or with circulating plasma levels of ACE2 may be explained by effects of these risk factors on susceptibility to severe COVID-19 through mechanisms unrelated to lung *ACE2* or *TMPRSS2* gene expression or plasma ACE2 protein expression.

Our study has a number of strengths. We used genetic variants as instrumental variables for studying the effect of ACEi drugs and cardiometabolic risk factors and were therefore able to investigate their causal effects on *ACE2* and *TMPRSS2* expression in the lung, and ACE levels in the plasma [32]. For ACEi drugs effects, we used two complementary instrument selection strategies based on associations of variants at the *ACE* locus with circulating serum ACE levels and SBP, respectively, and the consistent findings with both approaches add strength to our conclusions. Our Mendelian randomization approach is better able to overcome the confounding and reverse causation bias that can limit causal inferences from conventional epidemiological approaches [29,72]. Considering independent cohorts to assess lung expression of *ACE2* and *TMPRSS2* [44,45], and plasma levels of ACE2 [51], we were able to explore consistency in our results, and our conclusions are therefore less vulnerable to false-positive findings.

Our study also has limitations. We only investigated *ACE2* and *TMPRSS2* expression in the lung and circulating levels of ACE2 in the plasma, and it may be that expression in other tissues is more relevant to the risk and severity of COVID-19. Similarly, cellular ACE2 may have very different biological effects to circulating plasma ACE2. The GWAS analyses for lung *ACE2* expression and plasma ACE2 levels were all performed according to different protocols [44,45,51] and may therefore not be directly comparable. There was no available genetic instrument for the ARB antihypertensive drug class [35], and so we were not able to investigate this. The precision of our analyses was also limited, most notably for the lifetime smoking index results, which had widest confidence intervals. It may therefore be that our study was not sufficiently powered to exclude a clinically relevant effect for some exposures. The genetic variants that we used as instrumental variables may have pleiotropic effects where they affect the outcome through pathways independent of the exposure that they are proxying, and so bias the consequent Mendelian randomization estimates. While it is not possible to exclude this possibility, the relatively low heterogeneity detected between Mendelian randomization estimates produced by different variants, along with the consistency observed when performing analysis methods that are more robust to pleiotropy, suggests that this is unlikely to be a major source of bias [55]. A further reservation is the sample size available for genetic associations with the outcome measures, leading to limited power to detect a causal effect, particularly for the molecular outcomes. Finally, this study was not able to investigate off-target effects of ACE inhibitors that are unrelated to their intended protein target.

In summary, this Mendelian randomization study does not identify consistent evidence to support that ACEi antihypertensive drugs or cardiometabolic traits affect lung expression of *ACE2* and *TMPRSS2*, or plasma ACE2 levels. These findings therefore do not support a deviation from existing

expert consensus guidelines for the management of hypertension in the face of the current COVID-19 pandemic [60]. Efforts should be made by scientists and the news media to ensure that speculative stories with little evidential support are not propagated [73]. While cardiometabolic risk factors can be used to stratify patients in terms of their vulnerability to COVID-19, our data do not provide consistent support that the expression of *ACE2* or *TMPRSS2* represents causal mechanisms underlying these associations.

Ethics. The data used in this work are obtained from published studies that obtained relevant participant consent and ethical approval.

Data accessibility. All variants used as instruments and their genetic association estimates were selected from publicly available data sources and are provided in electronic supplementary material, Tables S1–S8. GWAS summary data for all the outcomes considered in this study are publicly available at http://dx.doi.org/10.6084/m9.figshare.12102681 (GTEx), http://dx.doi.org/10.6084/m9.figshare.12102711 (Lung eQTL Consortium), and http://dx.doi.org/10.6084/m9.figshare.12102777 (INTERVAL). The results from the analyses performed in this work are presented in the main manuscript or its supplementary files.

Authors' contributions. D.G. and S.B. designed the project. X.L., N.A.W., W.H.O., D.J.R., S.F., T.J., W.T., W.v.d.B., Y.B., P.J., K.H., A.B., D.D.S., J.E.P., A.S.B., B.E.E. and J.D. provided the data. S.B., D.G., J.E.P. and S.C.L. analysed the data. D.G., S.B., V.Z. and J.E.P. drafted the manuscript. All authors interpreted the results and critically revised the manuscript.

Competing interests. D.G. is a part-time employee of Novo Nordisk. A.S.B. reports grants outside of this work from AstraZeneca, Biogen, BioMarin, Bioverativ, Merck, Novartis, Pfizer and Sanofi and personal fees from Novartis. B.E.E. is on the scientific advisory boards of Freenome and Celsius Therapeutics and is a full-time employee at Genomics plc while on a leave of absence from Princeton University. J.E.P. has received travel and accommodation expenses and hospitality from Olink to speak at Olink-sponsored academic meetings. E.P. is a full-time employee of Pfizer. J.D. reports grants, personal fees and non-financial support from Merck Sharp & Dohme (MSD), grants, personal fees and non-financial support from Novartis, grants from Pfizer and grants from AstraZeneca outside the submitted work. J.D. sits on the International Cardiovascular and Metabolic Advisory Board for Novartis (since 2010); the Steering Committee of UK Biobank (since 2011); the MRC International Advisory Group (ING) member, London (since 2013); the MRC High Throughput Science 'Omics Panel Member, London (since 2013); the Scientific Advisory Committee for Sanofi (since 2013); the International Cardiovascular and Metabolism Research and Development Portfolio Committee for Novartis; and the AstraZeneca Genomics Advisory Board (since 2018). All other authors have no conflicts of interest to declare.

Funding. D.G. is supported by the Wellcome Trust 4i Programme (203928/Z/16/Z) and British Heart Foundation Centre of Research Excellence (RE/18/4/34215) at Imperial College London. S.B. is supported by a Sir Henry Dale Fellowship jointly funded by the Wellcome Trust and the Royal Society (award no. 204623/Z/16/Z). The academic coordinating centre for INTERVAL at Cambridge University was supported by core funding from: NIHR Blood and Transplant Research Unit in Donor Health and Genomics (NIHR BTRU-2014-10024), UK Medical Research Council (MR/L003120/1), British Heart Foundation (SP/09/002; RG/13/13/30194; RG/18/13/33946) and the NIHR (Cambridge Biomedical Research Centre at the Cambridge University Hospitals NHS Foundation Trust), and Health Data Research UK. S.F. is funded by the NIHR Blood and Transplant Research Unit in Donor Health and Genomics (NIHR BTRU-2014-10024). J.D. holds a British Heart Foundation Professorship and a National Institute for Health Research Senior Investigator Award. S.C.L. is funded by the Swedish Heart-Lung Foundation, the Swedish Research Council and the Swedish Research Council for Health, Working Life and Welfare. V.K. is funded by a European Union's Horizon 2020 research and innovation programme under the Marie Sklodowska-Curie grant no. 721567. V.Y.T. is supported by the Cancer Research UK (CRUK) Integrative Cancer Epidemiology Programme (C18281/A19169). S.M.L. is supported by an Academic Clinical Fellowship from the NIHR. Y.B. holds a Canada Research Chair in Genomics of Heart and Lung Diseases. J.E.P. is supported by a UKRI Innovation Fellowship at Health Data Research UK (MR/S004068/1). T.J. is funded by the National Institute for Health Research [Cambridge Biomedical Research Centre at the Cambridge University Hospitals NHS Foundation Trust]. A.M.M. is funded by the European Council Innovative Medicines Initiative (BigData@Heart). This work was supported by Health Data Research UK, which is funded by the UK Medical Research Council, Engineering and Physical Sciences Research Council, Economic and Social Research Council, Department of Health and Social Care (England), Chief Scientist Office of the Scottish Government Health and Social Care Directorates, Health and Social Care Research and Development Division (Welsh Government), Public Health Agency (Northern Ireland), British Heart Foundation and the Wellcome Trust. DNA extraction and genotyping was co-funded by the National Institute for Health Research (NIHR), the NIHR BioResource (http://bioresource.nihr.ac.uk) and the NIHR (Cambridge Biomedical Research Centre at the Cambridge University Hospitals NHS Foundation Trust). Donors were enrolled at Biospecimen Source Sites funded by Leidos Biomedical, Inc. (Leidos) subcontracts to the National Disease Research Interchange (10XS170) and Roswell Park Cancer Institute (10XS171). The Laboratory, Data Analysis and Coordinating Center (LDACC) was funded through a contract (HHSN268201000029C) to The Broad Institute, Inc. Biorepository operations were funded through a Leidos subcontract to Van Andel Institute (10ST1035). Additional data repository and project management provided by Leidos (HHSN261200800001E).

Acknowledgements. We would like to thank those children and family members who allowed us time to work on this manuscript despite disruptions to schooling and daily routine. Participants in the INTERVAL BioResource were

recruited with the active collaboration of NHS Blood and Transplant England (www.nhsbt.nhs.uk), which has supported fieldwork and other elements of the trial. The views expressed are those of the authors and not necessarily those of the NHS, the NIHR or the Department of Health and Social Care. A complete list of the investigators and contributors to the INTERVAL trial has previously been provided [Di Angelantonio E, Thompson SG, Kaptoge S, Moore C, Walker M, Armitage J *et al*. Efficiency and safety of varying the frequency of whole blood donation (INTERVAL): a randomized trial of 45 000 donors. *Lancet*. 2017;390(10110):2360-71.]. The academic coordinating centre would like to thank blood donor centre staff and blood donors for participating in the INTERVAL trial. The Genotype-Tissue Expression (GTEx) project was supported by the Common Fund of the Office of the Director of the National Institutes of Health (NIH). Additional funds from the National Cancer Institute; National Human Genome Research Institute (NHGRI); National Heart, Lung, and Blood Institute; National Institute on Drug Abuse; National Institute of Mental Health and National Institute of Neurological Disorders and Stroke. The authors would like to thank the staff at the IUCPQ Biobank of the Quebec Respiratory Heath Research Network for their valuable assistance with the lung eQTL dataset at Laval University.

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
