## [Reviewer comments · Royal Society Open Science]

Review History

RSOS-200958.R0 (Original submission)

Review form: Reviewer 1

Is the manuscript scientifically sound in its present form?

No

Are the interpretations and conclusions justified by the results?

No

Is the language acceptable?

Yes

Do you have any ethical concerns with this paper?

No

Have you any concerns about statistical analyses in this paper?

No

Recommendation?

Major revision is needed (please make suggestions in comments)

Comments to the Author(s)

Recent published commentaries have hypothesised adverse effects of increased ACE2 levels on severity of COVID-19 infection. Previously published studies suggest that ACE inhibitors and ARBs increase ACE2 levels. Therefore understanding the effects of these medications on ACE2 and TMPRSS2 levels, as well as the role of ACE2 and TMPRSS2 levels in Covid-19 severity, is of importance. This study aims to use MR analysis to investigate the former. My main comments are regarding the instrument selection for ACE and the interpretation of findings.

1) The authors state in their Abstract conclusion that “This study does not provide evidence to support that ACE inhibitor antihypertensive drugs affect lung ACE2 and TMPRSS2 expression or plasma ACE2 levels.” However, the MR analysis tests the causal effect of serum ACE levels on ACE2 or TMPRSS2 expression/levels, so a more accurate conclusion would be “This study does not provide evidence to support any effect of serum ACE levels (as a proxy for ACE inhibitors) on lung ACE2 and TMPRSS2 expression or plasma TMPRSS2 levels”. The authors need to acknowledge the caveat that they are unable to test unknown ‘off-target’ effects of different ACE inhibitors on ACE2 or TMPRSS2 levels.

2) Similarly, Pg 12 Line 11 reword ‘We did not find an association of genetically proxied ACE inhibition’ to ‘We did not find an association of lower serum ACE levels (as a proxy for ACE inhibition)’.

3) Pg 12 Line 12 of “These results therefore do not provide evidence to support that ACEi antihypertensive drugs affect risk or severity of COVID-19 through effects on ACE2 expression, as previously hypothesised”. This study does not specifically test this hypothesis, so to say “it does not provide evidence to support...” is not accurate. Please remove this sentence.

4) Related to the previous comment, GWAS summary data for Covid-19 severity are publicly available. It would be of interest to report results from an MR analysis of the effect of ACE, ACE2 and TMPRSS2 expression or plasma levels on COVID-19 severity.

5) Looking at the 17 variants associated with serum ACE levels, some of these are not very strongly associated with serum ACE levels e.g. rs117808108 which has an association p-value of $1E-3$ (F-statistic of 0.62). Apart from rs4343, which has an F-statistic of 64, all other variants have an F-statistic < 10 . Including such weak instruments in the main analysis is not justified. A more robust approach would be use only genome-wide significant SNPs in the main analysis.

6) The authors use rs4291 as a single instrument, because it is the only one associated with SBP. The rationale for this is not very clear. They show that SBP has no effect on ACE2/TMPRSS2 expression, which would imply that if there is any effect of ACE serum levels on ACE2/TMPRSS2 it would likely be independent of SBP. rs4291 is in LD with rs4343, the latter being the strongest associated SNP with serum ACE. rs4343 is also a proxy for the ACE indel and has been shown to be associated with plasma ACE activity (Chung et al PMID 20066004) and plasma ACE levels (Deming et al PMC4698720) at genome-wide significance, providing good justification for using rs4343 as a single SNP instrument rather than rs4291.

7) Were there any associations with serum ACE levels outside of the ACE locus (trans-associations)? If so sensitivity analyses should also include these.

8) Related to the previous question, two independent GWAS, one of plasma ACE levels (<https://doi.org/10.1038/srep18092>; GWAS summary data publicly available) and another of plasma ACE activity (<https://doi.org/10.1038/tpj.2009.70>) both identify variants near the ABO gene. It would be interesting to see if there is any difference in association between the cis and trans variants, especially since ABO blood groups have been implicated in Covid-19 severity, and most recently variants near ABO have been associated with severe COVID19 with respiratory failure (NEJM DOI: 10.1056/NEJMoa2020283).

Minor comments:

9) Could the authors provide brief details of the platform used to measure gene expression in the different eQTL studies.

10) For TMPRSS2 please provide brief details on how the two probe sets differ (e.g. do they measure different transcripts?). Also for figure 1, labelling as 'probe set 1' instead of 'consortium 1' would avoid any confusion.

11) In addition to the overall MR estimate, please provide supplementary plots showing the individual effects of each of the serum ACE SNP instruments on the following outcomes on the y-axis: SBP (as a positive control outcome), ACE2 expression, TMPRSS2 expression and plasma levels.

12) Could you provide the F-statistics for each ACE instrument in supplementary tables 1 and 2.

13) The authors show the effect of T2DM on ACE2 levels. It may also be worth looking at the effect of ACE2 levels on T2DM.

Review form: Reviewer 2 (Jane Zhao)

Is the manuscript scientifically sound in its present form?

Yes

Are the interpretations and conclusions justified by the results?

No

Is the language acceptable?

Yes

Do you have any ethical concerns with this paper?

No

Have you any concerns about statistical analyses in this paper?

Yes

Recommendation?

Major revision is needed (please make suggestions in comments)

Comments to the Author(s)

The study of Gill D et al. applied human genetic variants that proxy angiotensin-converting enzyme (ACE) inhibitor drug effects and cardiovascular risk factors to investigate whether these

exposures affect lung ACE2 and TMPRSS2 gene expression and circulating ACE2 levels. It is a very interesting study and has clinical implication. I have some suggestions as follows:

- 1) In the genetic variant selection, the authors used two sets of genetic instruments as proxy for ACEi. The first set is selected from SNPs in the ACE locus that were associated with serum ACE concentration in the Outcome Reduction with Initial Glargine INtervention (ORIGIN) trial and did not have strong pairwise correlation ($r^2 < 0.1$). I'm wondering whether the selected SNPs were also associated with blood pressure? Given that these SNPs were used to predict ACEi
- 2) In the selection of the genetic instrument for ACEi, an r^2 of 0.1 was used as cut-off, this seems inconsistent with the cutoff (0.001) used in selecting SNPs for cardiovascular risk factors. Why different cut-offs were used?
- 3) About the result on ACEi and ACE2 expression, given that GTEx and Lung eQTL Consortium are relatively small, with sample size of 515 and 1038 respectively, is it possible that the null association is due lack of power? This seems quite plausible for the association with ACE2 (Lung eQTL Consortium) in Figure 1.
- 4) The null association for smoking is confusing to me, especially when smoking is associated with severe COVID in a recent MR study (<https://europepmc.org/article/ppr/ppr178441>). Could the authors discuss more about this finding?

Decision letter (RSOS-200958.R0)

Dear Dr Burgess

The Editors assigned to your paper RSOS-200958 "ACE inhibition and cardiometabolic risk factors, lung ACE2 and TMPRSS2 gene expression, and plasma ACE2 levels: a Mendelian randomization study" have now received comments from reviewers and would like you to revise the paper in accordance with the reviewer comments and any comments from the Editors. Please note this decision does not guarantee eventual acceptance.

Both reviewers recognise the importance and interest of the work. However, both reviewers raise a number of substantive points regarding the analysis and the interpretation of the data. We invite you to respond to the comments supplied below and revise your manuscript. Below the referees' and Editors' comments (where applicable) we provide additional requirements. Final acceptance of your manuscript is dependent on these requirements being met. We provide guidance below to help you prepare your revision.

Please submit your revised manuscript and required files (see below) no later than 21 days from today's (ie 13-Oct-2020) date. Note: the ScholarOne system will 'lock' if submission of the revision is attempted 21 or more days after the deadline. If you do not think you will be able to meet this deadline please contact the editorial office immediately.

Please note article processing charges apply to papers accepted for publication in Royal Society Open Science (<https://royalsocietypublishing.org/rsos/charges>). Charges will also apply to papers transferred to the journal from other Royal Society Publishing journals, as well as papers

submitted as part of our collaboration with the Royal Society of Chemistry (<https://royalsocietypublishing.org/rsos/chemistry>). Fee waivers are available but must be requested when you submit your revision (<https://royalsocietypublishing.org/rsos/waivers>).

on behalf of Professor Mike Owen (Associate Editor) and Steve Brown (Subject Editor)
openscience@royalsociety.org

Associate Editor Comments to Author (Professor Mike Owen):

Both reviewers have raised issues around the analyses performed and their interpretation that need to be addressed before the paper is suitable for publication.

Reviewer comments to Author:

Reviewer: 1 Comments to the Author(s)

Recent published commentaries have hypothesised adverse effects of increased ACE2 levels on severity of COVID-19 infection. Previously published studies suggest that ACE inhibitors and ARBs increase ACE2 levels. Therefore understanding the effects of these medications on ACE2 and TMPRSS2 levels, as well as the role of ACE2 and TMPRSS2 levels in Covid-19 severity, is of importance. This study aims to use MR analysis to investigate the former. My main comments are regarding the instrument selection for ACE and the interpretation of findings.

1) The authors state in their Abstract conclusion that “This study does not provide evidence to support that ACE inhibitor antihypertensive drugs affect lung ACE2 and TMPRSS2 expression or plasma ACE2 levels.” However, the MR analysis tests the causal effect of serum ACE levels on ACE2 or TMPRSS2 expression/levels, so a more accurate conclusion would be “This study does not provide evidence to support any effect of serum ACE levels (as a proxy for ACE inhibitors) on lung ACE2 and TMPRSS2 expression or plasma TMPRSS2 levels”. The authors need to acknowledge the caveat that they are unable to test unknown ‘off-target’ effects of different ACE inhibitors on ACE2 or TMPRSS2 levels.

2) Similarly, Pg 12 Line 11 reword ‘We did not find an association of genetically proxied ACE inhibition’ to ‘We did not find an association of lower serum ACE levels (as a proxy for ACE inhibition)’.

3) Pg 12 Line 12 of “These results therefore do not provide evidence to support that ACEi antihypertensive drugs affect risk or severity of COVID-19 through effects on ACE2 expression, as previously hypothesised”. This study does not specifically test this hypothesis, so to say “it does not provide evidence to support...” is not accurate. Please remove this sentence.

4) Related to the previous comment, GWAS summary data for Covid-19 severity are publicly available. It would be of interest to report results from an MR analysis of the effect of ACE, ACE2 and TMPRSS2 expression or plasma levels on COVID-19 severity.

5) Looking at the 17 variants associated with serum ACE levels, some of these are not very strongly associated with serum ACE levels e.g. rs117808108 which has an association p-value of $1E-3$ (F-statistic of 0.62). Apart from rs4343, which has an F-statistic of 64, all other variants have an F-statistic < 10 . Including such weak instruments in the main analysis is not justified. A more robust approach would be use only genome-wide significant SNPs in the main analysis.

6) The authors use rs4291 as a single instrument, because it is the only one associated with SBP. The rationale for this is not very clear. They show that SBP has no effect on ACE2/TMPRSS2 expression, which would imply that if there is any effect of ACE serum levels on ACE2/TMPRSS2 it would likely be independent of SBP. rs4291 is in LD with rs4343, the latter being the strongest associated SNP with serum ACE. rs4343 is also a proxy for the ACE indel and has been shown to be associated with plasma ACE activity (Chung et al PMID 20066004) and plasma ACE levels (Deming et al PMC4698720) at genome-wide significance, providing good justification for using rs4343 as a single SNP instrument rather than rs4291.

7) Were there any associations with serum ACE levels outside of the ACE locus (trans-associations)? If so sensitivity analyses should also include these.

8) Related to the previous question, two independent GWAS, one of plasma ACE levels (<https://doi.org/10.1038/srep18092>; GWAS summary data publicly available) and another of plasma ACE activity (<https://doi.org/10.1038/tbj.2009.70>) both identify variants near the ABO gene. It would be interesting to see if there is any difference in association between the cis and trans variants, especially since ABO blood groups have been implicated in Covid-19 severity, and most recently variants near ABO have been associated with severe COVID19 with respiratory failure (NEJM DOI: 10.1056/NEJMoa2020283).

Minor comments:

9) Could the authors provide brief details of the platform used to measure gene expression in the different eQTL studies.

10) For TMPRSS2 please provide brief details on how the two probe sets differ (e.g. do they measure different transcripts?). Also for figure 1, labelling as 'probe set 1' instead of 'consortium 1' would avoid any confusion.

11) In addition to the overall MR estimate, please provide supplementary plots showing the individual effects of each of the serum ACE SNP instruments on the following outcomes on the y-axis: SBP (as a positive control outcome), ACE2 expression, TMPRSS2 expression and plasma levels.

12) Could you provide the F-statistics for each ACE instrument in supplementary tables 1 and 2.

13) The authors show the effect of T2DM on ACE2 levels. It may also be worth looking at the effect of ACE2 levels on T2DM.

Reviewer: 2

Comments to the Author(s)

The study of Gill D et al. applied human genetic variants that proxy angiotensin-converting enzyme (ACE) inhibitor drug effects and cardiovascular risk factors to investigate whether these exposures affect lung ACE2 and TMPRSS2 gene expression and circulating ACE2 levels. It is a very interesting study and has clinical implication. I have some suggestions as follows:

1) In the genetic variant selection, the authors used two sets of genetic instruments as proxy for ACEi. The first set is selected from SNPs in the ACE locus that were associated with serum ACE

concentration in the Outcome Reduction with Initial Glargine INtervention (ORIGIN) trial and did not have strong pairwise correlation ($r^2 < 0.1$). I'm wondering whether the selected SNPs were also associated with blood pressure? Given that these SNPs were used to predict ACEi

2) In the selection of the genetic instrument for ACEi, an r^2 of 0.1 was used as cut-off, this seems inconsistent with the cutoff (0.001) used in selecting SNPs for cardiovascular risk factors. Why different cut-offs were used?

3) About the result on ACEi and ACE2 expression, given that GTEx and Lung eQTL Consortium are relatively small, with sample size of 515 and 1038 respectively, is it possible that the null association is due lack of power? This seems quite plausible for the association with ACE2 (Lung eQTL Consortium) in Figure 1.

4) The null association for smoking is confusing to me, especially when smoking is associated with severe COVID in a recent MR study (<https://europepmc.org/article/ppr/ppr178441>). Could the authors discuss more about this finding?

===PREPARING YOUR MANUSCRIPT===

- one version identifying all the changes that have been made (for instance, in coloured highlight, in bold text, or tracked changes);
- a 'clean' version of the new manuscript that incorporates the changes made, but does not highlight them. This version will be used for typesetting if your manuscript is accepted.

===PREPARING YOUR REVISION IN SCHOLARONE===

Author's Response to Decision Letter for (RSOS-200958.R0)

See Appendix A.

Decision letter (RSOS-200958.R1)

Dear Dr Burgess,

It is a pleasure to accept your manuscript entitled "ACE inhibition and cardiometabolic risk factors, lung ACE2 and TMPRSS2 gene expression, and plasma ACE2 levels: a Mendelian randomization study" in its current form for publication in Royal Society Open Science. The comments of the editor who considered your revisions are included at the foot of this letter.

COVID-19 rapid publication process:

We are taking steps to expedite the publication of research relevant to the pandemic. If you wish, you can opt to have your paper published as soon as it is ready, rather than waiting for it to be published the scheduled Wednesday.

This means your paper will not be included in the weekly media round-up which the Society sends to journalists ahead of publication. However, it will still appear in the COVID-19 Publishing Collection which journalists will be directed to each week (<https://royalsocietypublishing.org/topic/special-collections/novel-coronavirus-outbreak>).

If you wish to have your paper considered for immediate publication, or to discuss further, please notify openscience_proofs@royalsociety.org and press@royalsociety.org when you respond to this email.

Please note that we require active email addresses for all authors on the paper. At present, the following are showing as inactive or not receiving messages - please can you confirm these are correct addresses or provide alternatives:

marvanitis@jhu.edu;

asaha@jhu.edu;

a.lunt@imperial.ac.uk

Kind regards,

Andrew Dunn

on behalf of Professor Mike Owen (Associate Editor) and Steve Brown (Subject Editor)
openscience@royalsociety.org

Associate Editor Comments to Author (Professor Mike Owen):

Associate Editor

Comments to the Author:

I am pleased to accept your paper for publication.

Appendix A

Imperial College
London

UNIVERSITY OF
CAMBRIDGE

October 23, 2020

Dear Professor Owen and Professor Brown,

We thank you for the careful assessment of our original research article, 'ACE inhibition and cardiometabolic risk factors, lung *ACE2* and *TMPRSS2* gene expression, and plasma *ACE2* levels: a Mendelian randomization study', and the constructive feedback provided by the reviewers. We have now revised the manuscript accordingly, providing detailed responses to the reviewer comments below, and find the work to be markedly improved as a result.

We look forward to your decision.

Best wishes,

Dipender Gill MBCh PhD

Physician Scientist, Department of Clinical Pharmacology and Therapeutics, Imperial College Healthcare NHS Trust

Stephen Burgess PhD

Group Leader, MRC Biostatistics Unit and Cardiovascular Epidemiology Unit, University of Cambridge

Associate Editor Comments to Author (Professor Mike Owen):

Both reviewers have raised issues around the analyses performed and their interpretation that need to be addressed before the paper is suitable for publication.

We thank the associate editor for his comments and assessment of the paper. We note that in revision, we realized that the genetic association with serum ACE levels was incorrectly aligned for one of the genetic variants. We have corrected this, and have revised estimates accordingly. A consequence of this is that genetically predicted serum ACE levels are associated with gene expression of *ACE2* in the Lung eQTL dataset at a nominal significance level, although this result is not replicated in further analyses. We have updated the manuscript to reflect this.

Reviewer 1

Recent published commentaries have hypothesised adverse effects of increased ACE2 levels on severity of COVID-19 infection. Previously published studies suggest that ACE inhibitors and ARBs increase ACE2 levels. Therefore understanding the effects of these medications on ACE2 and TMPRSS2 levels, as well as the role of ACE2 and TMPRSS2 levels in Covid-19 severity, is of importance. This study aims to use MR analysis to investigate the former. My main comments are regarding the instrument selection for ACE and the interpretation of findings.

We thank the reviewer for their careful assessment of our work, and the constructive feedback provided.

1) The authors state in their Abstract conclusion that “This study does not provide evidence to support that ACE inhibitor antihypertensive drugs affect lung ACE2 and TMPRSS2 expression or plasma ACE2 levels.” However, the MR analysis tests the causal effect of serum ACE levels on ACE2 or TMPRSS2 expression/levels, so a more accurate conclusion would be “This study does not provide evidence to support any effect of serum ACE levels (as a proxy for ACE inhibitors) on lung ACE2 and TMPRSS2 expression or plasma TMPRSS2 levels”. The authors need to acknowledge the caveat that they are unable to test unknown ‘off-target’ effects of different ACE inhibitors on ACE2 or TMPRSS2 levels.

We acknowledge the reviewer’s point and have made the suggested change in the Abstract. We have also added this as a limitation in the Discussion section: “This study was not able to investigate off-target effects of ACE inhibitors that are unrelated to their intended protein target”.

2) Similarly, Pg 12 Line 11 reword ‘We did not find an association of genetically proxied ACE inhibition’ to ‘We did not find an association of lower serum ACE levels (as a proxy for ACE inhibition)’.

We thank the reviewer for highlighting this, and have incorporated the suggestion.

3) Pg 12 Line 12 of “These results therefore do not provide evidence to support that ACEi antihypertensive drugs affect risk or severity of COVID-19 through effects on ACE2 expression, as previously hypothesised”. This study does not specifically test this hypothesis, so to say “it does not provide evidence to support...”

is not accurate. Please remove this sentence.

This sentence has now been removed.

4) Related to the previous comment, GWAS summary data for Covid-19 severity are publicly available. It would be of interest to report results from an MR analysis of the effect of ACE, ACE2 and TMPRSS2 expression or plasma levels on COVID-19 severity.

We have added analyses in which we investigate the associations between genetically predicted serum ACE levels and risk of hospitalization due to COVID-19, taking genetic associations with COVID-19 hospitalization versus population-based controls from the COVID-19 Host Genomics Initiative (release 4 alpha). The reason for this choice of outcome is that associations with COVID-19 hospitalization are less susceptible to selection bias than analyses for COVID-19 diagnosis, as testing regimes typically do not test all individuals equally.

While the idea of including analyses considering the effects of *ACE2* and *TMPRSS2* expression are appealing, it is currently beyond the scope of the paper. Additionally, the *ACE2* coding gene region is on the X chromosome, and genetic associations for variants on the X chromosome are typically not published by large consortia. We are unaware of common variants in the *TMPRSS2* gene region that explain sufficient variance in *TMPRSS2* gene expression that could be used in a Mendelian randomization investigation.

5) Looking at the 17 variants associated with serum ACE levels, some of these are not very strongly associated with serum ACE levels e.g. rs117808108 which has an association p-value of 1E-3 (F-statistic of 0.62). Apart from rs4343, which has an F-statistic of 64, all other variants have an F-statistic < 10. Including such weak instruments in the main analysis is not justified. A more robust approach would be use only genome-wide significant SNPs in the main analysis.

While a genome-wide significance threshold is necessary when performing a genome-wide association study, in which association tests are performed for hundreds of thousands of variants from across the genome, in this case only a single gene region is considered. It is unlikely that any of these variants are associated with serum ACE levels due to chance alone (false positives), even when the p-value is only $\sim 10^{-3}$. Hence, it is unnecessary to be conservative in selecting genetic variants, particularly as we want to maximize the variance on the risk factor explained by the genetic variants. As an aside, a genetic variant with a p-value of 0.001 will have an F statistic of 10 in the dataset under analysis. However, in this case the individual contributions of variants to the F statistic is less important than their combined contribution.

We have maintained the current choice of variants in the main analysis, however as per the reviewer's suggestion we have also provided supplementary analyses restricted to variants associated with serum ACE levels at $p < 5 \times 10^{-8}$. Results are substantially unchanged (Supplementary Figure 2a):

6) The authors use rs4291 as a single instrument, because it is the only one associated with SBP. The rationale for this is not very clear. They show that SBP has no effect on ACE2/TMPRSS2 expression, which would imply that if there is any effect of ACE serum levels on ACE2/TMPRSS2 it would likely be independent of SBP. rs4291 is in LD with rs4343, the latter being the strongest associated SNP with serum ACE. rs4343 is also a proxy for the ACE indel and has been shown to be associated with plasma ACE activity (Chung et al PMID 20066004) and plasma ACE levels (Deming et al PMC4698720) at genome-wide significance, providing good justification for using rs4343 as a single SNP instrument rather than rs4291.

The rs4343 variant is included in the primary analyses as one of the variants that predict serum ACE levels. Hence analyses including this variant were already presented in the initial submission. In response to the reviewer’s suggestion, we have added supplementary analyses using the rs4343 variant only; results are substantially unchanged (Supplementary Figure 2b; result not available for the GTEx consortium because the rs4343 variant is missing):

7) Were there any associations with serum ACE levels outside of the ACE locus (trans-associations)? If so sensitivity analyses should also include these.

Generally speaking, Mendelian randomization analyses are more reliable when they focus on *cis* variants (that is, variants in the coding region for the relevant risk factor) where this is possible. It is rare that an investigator can state with confidence that the instrumental variable assumptions are satisfied for a *trans* variant.

In this specific case, the only *trans* variants that have been shown to be associated with serum ACE levels are in the *ABO* gene region (see references in this reviewer's point 8). This gene region is known to be highly pleiotropic, and variants in this region are associated with a large number of traits. Hence, there is little incentive to include *trans* variants in our analyses, as this would likely generate misleading results related to pleiotropic associations unrelated to ACE.

8) Related to the previous question, two independent GWAS, one of plasma ACE levels (<https://doi.org/10.1038/srep18092>; GWAS summary data publicly available) and another of plasma ACE activity (<https://doi.org/10.1038/tpj.2009.70>) both identify variants near the ABO gene. It would be interesting to see if there is any difference in association between the *cis* and *trans* variants, especially since ABO blood groups have been implicated in Covid-19 severity, and most recently variants near ABO have been associated with severe COVID19 with respiratory failure (NEJM DOI: 10.1056/NEJMoa2020283).

As stated above, variants in the *ABO* locus are highly pleiotropic and hence not suitable for use in a Mendelian randomization investigation.

Minor comments:

9) Could the authors provide brief details of the platform used to measure gene expression in the different eQTL studies.

We have now added these details to the Methods section. For the GTEx project we write, 'RNA sequencing was performed using the Illumina TruSeq™ RNA sample preparation protocol and gene-level expression quantification was performed using RNA-SeQC for gene-level read counts and Transcripts per Million values (42)'. For the Lung eQTL Consortium we write, 'Expression profiling was performed using an Affymetrix custom array (see GEO platform GPL10379) (40)'.

10) For TMPRSS2 please provide brief details on how the two probe sets differ (e.g. do they measure different transcripts?). Also for figure 1, labelling as 'probe set 1' instead of 'consortium 1' would avoid any confusion.

The probe sets do measure different transcripts as illustrated in the figure provided. The green rectangle corresponds to probe set 1 and all probes are located in exon 14. The red rectangle is for probe set 2 and only two probes are located in exon 2 of a specific transcript (AK313338).

We have added to the Methods section that the probe sets measure different transcripts. The specific probes for the two *TMPRSS2* probe sets are now detailed in Supplementary Table 1.

Supplementary Table 1. The specific probes for the two *TMPRSS2* probe sets.

Probe set	Probe sequence	Binding
Probe set 1 (100130004_TGI_at)	GTTTTGTTTTGGACTCTCTGTGGTC	Exon 14 TMPRSS2
	TTTGTCTTGGACTCTCTGTGGTCCC	Exon 14 TMPRSS2
	GCTTTGACAAATGACTGGCTCCTG	Exon 14 TMPRSS2
	TTGCCAAGTAAGAGTGGTGGCCTAT	Exon 14 TMPRSS2
	TGCCAAGTAAGAGTGGTGGCCTATT	Exon 14 TMPRSS2
	TTTGACAAATGACTGGCTCCTGAC	Exon 14 TMPRSS2
	TCACCTTTGCAAGTAAGAGTGGTG	Exon 14 TMPRSS2
	TTTTGTTTGGACTCTCTGTGGTCC	Exon 14 TMPRSS2
Probe set 2 (100157336_TGI_at)	GTCCCCTGCTACAGGGCATTGAGGT	Exon 2 AK313338
	CCACCTCCCCTTAGAGAATATTT	intronic

	AGTTTCGTA ACTCCTGCCGCATAGT	intronic
	CAGAGCTTTGAGAAGGCTGTTATCA	intronic
	ATTGAGGTGAGGTCCGCCTTTGCC	intronic
	TGTTTCTTTTTGAACTTGCCACCTC	intronic
	GAGAGCGCTGCTTTCAGAGCTTTGA	intronic
	AGAGAGTGACCCGTGCATCTTTCCA	Exon 2 AK313338
	TGCCGCATAGTTGGTGCCTGCTCTC	intronic

By way of interpretation, we have added: “Expression levels were much higher for the first probe set, and so these results are more reliable.

11) In addition to the overall MR estimate, please provide supplementary plots showing the individual effects of each of the serum ACE SNP instruments on the following outcomes on the y-axis: SBP (as a positive control outcome), ACE2 expression, TMPRSS2 expression and plasma levels.

These figures are now presented as Supplementary Figure 1.

12) Could you provide the F-statistics for each ACE instrument in supplementary tables 1 and 2.

As the genetic variants are correlated, it does not make sense to provide F statistics for each variant individually. For example, if two variants are highly correlated, then they may individually have high F statistics, but they do not explain independent variation in the risk factor. Additionally, the F statistic depends on the sample size, but the relevant sample size is for the outcome dataset. Hence F statistics differ between the analyses presented. We have therefore provided overall F statistics for the sets of variants in the manuscript text. In the first paragraph of the Methods section, we write, “Accounting for correlation, these variants explain 29.0% of the variance in serum ACE concentration, corresponding to an F statistic of 85.4 (INTERVAL), 17.7 (Lung eQTL Consortium), and 8.8 (GTEX).”

13) The authors show the effect of T2DM on ACE2 levels. It may also be worth looking at the effect of ACE2 levels on T2DM.

As per the response to point 4, this is outside the current scope of the work. Furthermore, such analysis would require genetic associations with type 2 diabetes for variants in the ACE2 gene region, which is on the X chromosome. Consortium data on associations with type 2 diabetes for these variants are unfortunately not publicly available.

Reviewer 2

The study of Gill D et al. applied human genetic variants that proxy angiotensin-converting enzyme (ACE) inhibitor drug effects and cardiovascular risk factors to investigate whether these exposures affect lung *ACE2* and *TMPRSS2* gene expression and circulating ACE2 levels. It is a very interesting study and has clinical implication.

We thank the reviewer for their insightful and helpful comments.

I have some suggestions as follows:

1) In the genetic variant selection, the authors used two sets of genetic instruments as proxy for ACEi. The first set is selected from SNPs in the ACE locus that were associated with serum ACE concentration in the Outcome Reduction with Initial Glargine Intervention (ORIGIN) trial and did not have strong pairwise correlation ($r^2 < 0.1$). I'm wondering whether the selected SNPs were also associated with blood pressure? Given that these SNPs were used to predict ACEi

As per reviewer 1 point 11, we have now plotted the genetic associations with serum ACE level versus the genetic association with systolic blood pressure (Supplementary Figure 1a). We have added to the Results section: "The variants were associated with SBP in the expected direction: 0.22 mmHg (95% confidence interval 0.06 to 0.37, $p=0.006$) increase per 1 standard deviation increase in plasma ACE."

2) In the selection of the genetic instrument for ACEi, an r^2 of 0.1 was used as cut-off, this seems inconsistent with the cutoff (0.001) used in selecting SNPs for cardiovascular risk factors. Why different cut-offs were used?

The reason for the different choice of variants is that analyses for serum ACE levels include variants from a single gene region, whereas analyses for the cardiovascular risk factors included variants throughout the genome. For serum ACE levels, we only select variants from a single gene region, so selecting correlated variants is necessary to help obtain sufficient power. However, we did not suffer this issue for the cardiovascular risk factors, and so were able to adopt a more stringent correlation threshold.

3) About the result on ACEi and ACE2 expression, given that GTEx and Lung eQTL Consortium are relatively small, with sample size of 515 and 1038 respectively, is it possible that the null association is due lack of power? This seems quite plausible for the association with ACE2 (Lung eQTL Consortium) in Figure 1.

The reviewer makes a valid point that we now acknowledge in the limitations section as follows: "A further reservation is the sample size available for genetic associations with the outcome measures, leading to limited power to detect a causal effect, particularly for the molecular outcomes."

4) The null association for smoking is confusing to me, especially when smoking is associated with severe COVID in a recent MR study (<https://europepmc.org/article/ppr/ppr178441>). Could the authors discuss more about this finding?

The reviewer highlights an interesting observation. It may be that both body mass index and smoking are affecting COVID-19 susceptibility through pathways other than *ACE2* and *TMPRSS2* expression. We now detail this in the Discussion section as follows: “Since initial submission of this manuscript, further data have become available on genetic associations with susceptibility to COVID-19 (41, 65). Mendelian randomization analyses have supported an effect of higher BMI and lifetime smoking on increasing susceptibility to severe COVID-19 (66). The discrepancy with our current findings, which did not identify an association of genetically predicted BMI or smoking with lung *ACE2* and *TMPRSS2* expression or with circulating plasma levels of *ACE2* may be explained by effects of these risk factors on susceptibility to severe COVID-19 through mechanisms unrelated to lung *ACE2* or *TMPRSS2* gene expression or plasma *ACE2* protein expression.”